# Effect of CuO Loading on the Photocatalytic Activity of SrTiO₃ for Hydrogen Evolution

**Xuan Truong Mai [1], Duc Nguyen Bui [1,\*], Van Khang Pham [1] , Thi Hien Lan Nguyen [1], Thi To Loan Nguyen [1] , Hung Dung Chau [2] and Thi Kim Ngan Tran [2,\*]**

[1] Faculty of Chemistry, Thai Nguyen University of Education, No 20 Luong Ngoc Quyen Street, Thai Nguyen City 24000, Vietnam

[2] Institute of Applied Technology and Sustainable Development, Nguyen Tat Thanh University, Ho Chi Minh City 700000, Vietnam

\* Correspondence: ducnguyen@tnue.edu.vn (D.N.B.); nganttk@ntt.edu.vn (T.K.N.T.)

**Abstract:** A CuO-loaded SrTiO₃ catalyst showed highly photocatalytic activity for H₂ evolution. This catalyst was prepared by an impregnation method and characterized by XRD, TEM, BET, XPS, Uv-vis DRS and PL techniques. Under optimum conditions, the best rate of H₂ evolution of the CuO-loaded SrTiO₃ catalyst is 5811 μmol h⁻¹g⁻¹, whereas it is a mere 34 μmol h⁻¹g⁻¹ for the pure SrTiO₃. High efficiency, low cost and good stability are some of the merits that underline the promising potential of CuO-loaded SrTiO₃ in the photocatalytic hydrogen.

**Keywords:** strontium titanate (SrTiO₃); co-catalyst; photocatalytic; hydrogen evolution

## 1. Introduction

It is well known that photocatalytic hydrogen evolution from water splitting has received much attention in recent years for its potential application in providing hydrogen as a clean and renewable energy resource, even on a large scale [1–6]. Therefore, to find an efficient photocatalyst for hydrogen evolution is a vital topic.

Strontium titanate (SrTiO₃) has typical perovskite structure with the advantages of low cost and excellent chemical stability. It has been widely used as a photocatalyst [7–9]. Although perovskite-type SrTiO₃ appears to be a promising candidate for photocatalytic hydrogen evolution from water splitting since its suitable band structure for facilitating hydrogen and oxygen formation [10], the photocatalytic efficiency of pure SrTiO₃ for hydrogen evolution is still very low, mainly due to the fast recombination of the photo-generated electrons and holes. It is necessary to modify the SrTiO₃ particle to obtain an active photocatalyst for water splitting.

Up to now, many outstanding methods have been developed to improve the photocatalytic efficiency of SrTiO₃ semiconductor for hydrogen evolution, such as doped with metals or non-metals [11–18], or coupled with other semiconductors [19–27], have been investigated to solve the above mentioned issues.

It is known that CuO can act as an efficient cocatalyst of TiO₂ for its photocatalytic H₂ production. The band gap energy (E₉) of SrTiO₃ and TiO₂ is similar (3.2 eV), and it is higher than that of CuO (1.7 eV). Therefore, it can be expected that the CuO may be an efficient cocatalyst for SrTiO₃, which is meaningful for us to prepare a highly efficient, cheap and stable SrTiO₃-based photocatalyst. In fact, several investigators have been focused on the visible photocatalysts based on SrTiO₃ and CuO. Choudhary et al. have prepared CuO/SrTiO₃ bilayered thin films by sol–gel spin-coating technique, which were used for a water splitting reaction. According to the literature, the bilayered system offered enhanced photoconversion efficiency, attributed to improved conductivity, which ameliorate separation of the photo-generated carriers at the CuO/STO interface and higher value of flatband potential [28]. Recently, Ahmadi et al. [29] synthesized CuO/SrTiO₃

nanoparticles using a combination of impregnation and precipitation-deposition method, and the photocatalytic activity of CuO/SrTiO$_3$ nanoparticles was evaluated by degradation of RhB under UV light irradiation. According to the mechanism in the literature, the CuO could help to separate the photo-generated electron-hole efficiently. Sepideh et al. [30] synthesized CuO/SrTiO$_3$ composites. The prepared materials were studied as photocatalysts for the hydrogen evolution from aqueous methanol solution at an ambient temperature under UV light irradiation. In comparison to unmodified SrTiO$_3$, the highest increase in photocatalytic activity was more than threefold, i.e., from 39 to 130 µmol h$^{-1}$. According to the literature, the improvement of the photocatalytic activity of SrTiO3 for hydrogen evolution from aqueous methanol solution is attributed to the combination with CuO as a narrow band gap semiconductor to obtain CuO/SrTiO$_3$ composites.

In the present work, we will explore the effect of CuO as a cocatalyst on the photocatalytic activity of the SrTiO$_3$ photocatalyst for hydrogen evolution from aqueous solution containing various electron donors. The prepared CuO-SrTiO$_3$ photocatalysts were characterized by XRD, TEM, EDX and XPS analysis. The DSR spectra were used to explore whether or not the effect of CuO on the band gap energy of the SrTiO$_3$ semiconductor and the PL emission spectra were used to reveal the efficiency of trapping, transfer and separation of charge carriers, and to investigate their lifetime in the semiconductors. Moreover, the effects of some factors, such as type and concentration of electron donors, reaction temperature and photocatalyst concentration on the hydrogen evolution rate, will be systematically investigated in detailed. In addition, for comparison, the photocatalytic hydrogen evolution over Pt-SrTiO$_3$ was also carried out under the same conditions with the case of CuO-SrTiO$_3$ samples.

## 2. Results

### 2.1. Characterization of Photocatalyst

The XRD patterns of the pure SrTiO$_3$ and the 1.5 wt.% CuO-loaded SrTiO$_3$ are shown in Figure 1.

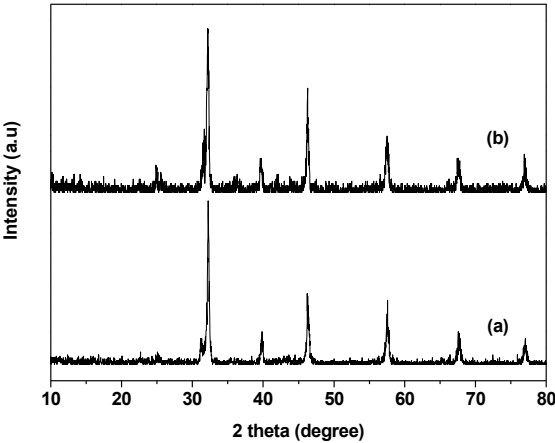

**Figure 1.** XRD patterns of pure SrTiO$_3$ (a) and 1.5 wt.% CuO-loaded SrTiO$_3$ (b).

The result in Figure 1 shows that the X-ray diffraction pattern of the 1.5 wt.% CuO-SrTiO$_3$ sample almost coincides with that of the pure SrTiO$_3$. Diffraction peaks at 32.42°, 39.98°, 46.48°, 57.79°, 67.80° and 77.17° in all of XRD patterns are corresponded to the (110), (111), (200), (211), (220) and (310) planes of cubic SrTiO$_3$ (JCPDS card No. 35-0734) [31], respectively. This result indicates that the obtained SrTiO$_3$ possesses pure cubic phase. The phase of CuO cocatalyst is not observed in the XRD pattern of the 1.5 wt.% CuO-SrTiO$_3$. It is possibly due to the concentration of the CuO cocatalyst in the photocatalyst is lower than that afforded by XRD detection limits or high dispersion of CuO on the SrTiO$_3$ surface. The experimental result below also confirms this deduction.

As shown in TEM images of Figure 2, the introduction of CuO does not obviously change the morphology of SrTiO$_3$ nanoparticles. The mean diameter of the CuO-loaded

SrTiO$_3$ is approximately 35 nm, and the surface of the CuO-loaded SrTiO$_3$ nanoparticles is fairly smooth. It is indicated that the CuO particles are homogenously distributed through the surface of SrTiO$_3$ supports. It is consistent with the result of XRD analysis.

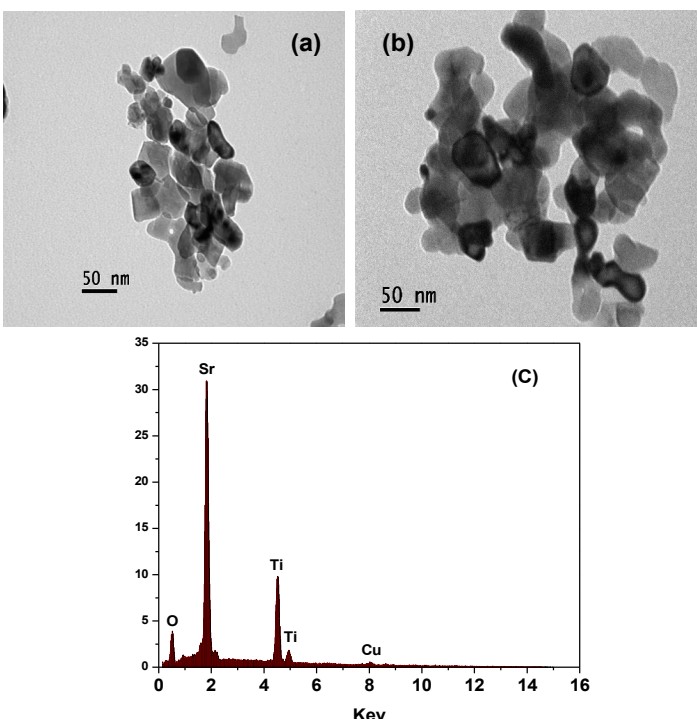

**Figure 2.** TEM images of SrTiO$_3$ (**a**), 1.5% CuO-SrTiO$_3$ (**b**) and EDX spectra of 1.5% CuO-SrTiO$_3$ (**c**).

The evidence for the composition of 1.5% CuO-SrTiO$_3$ was obtained by EDX analysis as shown in Figure 2c. The result of EDX shows that the product is composed of the elements Sr, Ti, O and Cu. Further, the XPS analysis of 1.5% CuO-SrTiO$_3$ sample was performed, and the survey spectrum and high-resolution scans are shown in Figure 3.

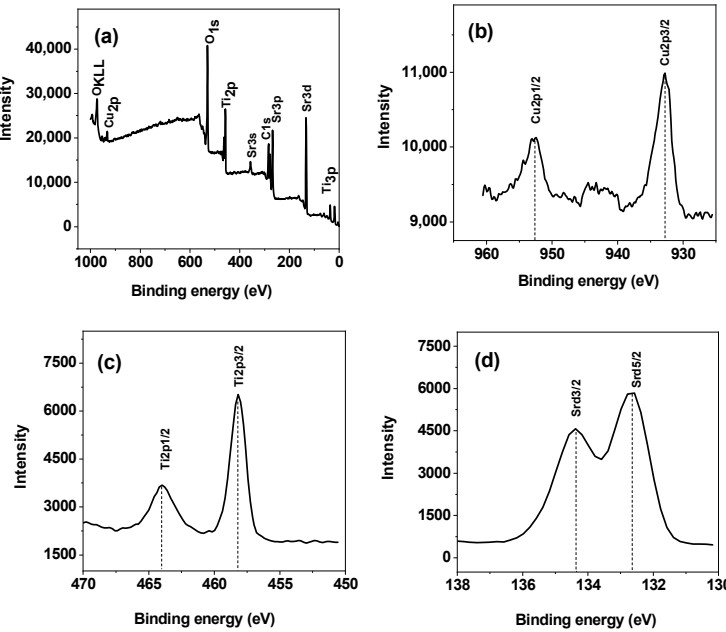

**Figure 3.** XPS spectra of the samples: (**a**) XPS survey spectrum of 1.5% CuO-SrTiO$_3$ sample; (**b**) high-resolution Cu 2p spectrum; (**c**) high-resolution Ti 2p spectrum and (**d**) high-resolution Sr d spectrum.

From the XPS survey spectrum shown in Figure 3a, Cu, Sr, Ti and O photoelectron lines from 1.5% CuO-SrTiO$_3$ sample are detected along with C peak. It is consistent with EDX results. It should be noted that the carbon peak (C 1 s) in the survey spectra is attributed to the residual carbon from the sample and trace hydrocarbon from XPS instrument itself [32]. In Figure 3b, spin orbital splitting photoelectrons of Cu 2p$_{3/2}$ and Cu 2p$_{1/2}$ are located at 933.6 eV and 952.6 eV, respectively, which correspond to CuO [33,34]. A shake-up line among the binding energies of Cu 2p$_{3/2}$ and Cu 2p$_{1/2}$ further confirms this oxidation state, which is in a good agreement with the results of the literature reported. As shown in Figure 3c, the respective binding energies of Ti 2p$_{3/2}$ and Ti 2p$_{1/2}$ are located at 458.2 and 464.0 eV. The two bands are assigned to typical Ti$^{4+}$ [35]. In Figure 3d, the binding energies of Sr d$_{5/2}$ and Sr d$_{3/2}$ at 133.5 and 134.4 eV, respectively, are assigned to typical Sr$^{2+}$ [31]. The result of XPS analysis demonstrates that the Cu element in SrTiO$_3$ is in the form of CuO.

### 2.2. Effect of Loading Amount of CuO on Hydrogen Evolution Activity over SrTiO$_3$

In order to evaluate the influence of loading CuO on the photocatalytic activity for hydrogen evolution over SrTiO$_3$ photocatalysts, the photocatalytic activity for hydrogen evolution is conducted over pure SrTiO$_3$, the CuO-SrTiO$_3$ photocatalysts and Pt-SrTiO$_3$ photocatalysts under the same conditions, respectively. The amount of hydrogen evolution for the first 2 h is used as a comparative indicator of the hydrogen evolution activity. The experimental results are shown in Figure 4 and Table 1.

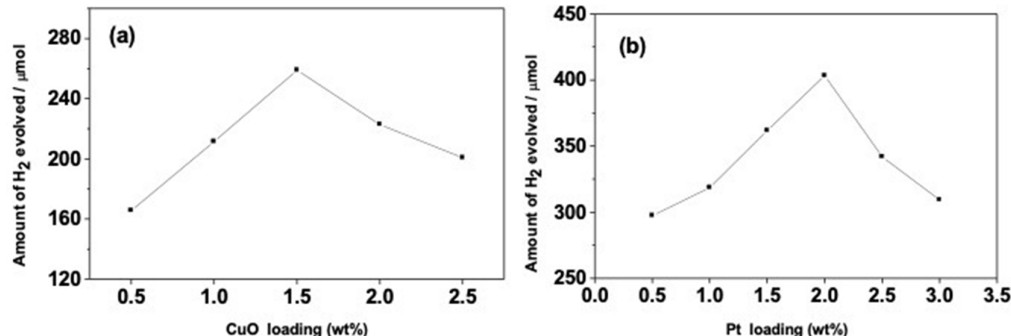

**Figure 4.** The effects of the loading amount of CuO (**a**) and Pt (**b**) on the photocatalytic hydrogen evolution activity of SrTiO$_3$ (catalyst: 60 mg; methanol content: 60 mL, 60 vol.%; reaction temperature: 25 °C).

**Table 1.** The photocatalytic hydrogen evolution over various photocatalysts (photocatalyst: 60 mg; the content of methanol: 60 mL, 60 vol.%; temperature: 25 °C; irradiation time: 2 h).

| Photocatalysts | Amount of Hydrogen Evolution (μmol) |
| :---: | :---: |
| Pure SrTiO$_3$ | 1.45 |
| 0.5 wt.% CuO-loaded SrTiO$_3$ | 165 |
| 1.0 wt.% CuO-loaded SrTiO$_3$ | 211 |
| 1.5 wt.% CuO-loaded SrTiO$_3$ | 258 |
| 2.0 wt.% CuO-loaded SrTiO$_3$ | 222 |
| 2.5 wt.% CuO-loaded SrTiO$_3$ | 200 |
| 0.5 wt.% Pt-loaded SrTiO$_3$ | 297 |
| 1.0 wt.% Pt-loaded SrTiO$_3$ | 318 |
| 1.5 wt.% Pt-loaded SrTiO$_3$ | 361 |
| 2.0 wt.% Pt-loaded SrTiO$_3$ | 403 |
| 2.5 wt.% Pt-loaded SrTiO$_3$ | 341 |
| 3.0 wt.% Pt-loaded SrTiO$_3$ | 309 |

It is clear that the amount of hydrogen evolution over pure SrTiO$_3$ is only 1.45 μmol. While the photocatalytic activity of the SrTiO$_3$ photocatalyst for hydrogen evolution can be greatly enhanced by loading CuO cocatalyst. The maximum amounts of hydrogen

evolution from 60 vol.% methanol aqueous solution over CuO-SrTiO$_3$ (1.5 wt.% loaded) and Pt-SrTiO$_3$ (2.0 wt.% loaded) are 259 μmol and 403 μmol, respectively. Although the photocatalytic activity of the CuO-SrTiO$_3$ is lower than that of Pt-SrTiO$_3$, its rate of H$_2$ evolution is still more than 60% of that of SrTiO$_3$ nanoparticles with optimum loading amount of Pt.

It can be seen that the CuO cocatalyst play critical roles in improving both the activity and stability of SrTiO$_3$ photocatalysts in photocatalytic processes. Firstly, CuO can be contribute to the electron–hole separation at the CuO/SrTiO$_3$ interface because, in the case of the SrTiO$_3$ photocatalyst loaded with CuO cocatalyst, the photogenerated electrons can migrate to the surface of cocatalyst and the photogenerated holes are trapped in the SrTiO$_3$ surface. Thus, the recombination rate of e$^-$/h$^+$ pairs is decreased, and the rate of hydrogen evolution is increased. Secondly, CuO cocatalyst could decrease the activation energy or overpotential for redox reactions such as H$_2$ and O$_2$ evolution. Thirdly, CuO cocatalyst can inhibit photo-corrosion and improve the stability of semiconductor photocatalysts. In photocatalytic reactions, there are a number of visible-light-responsive semiconductors that are likely oxidized by photogenerated holes and causing their self-decomposition. The cocatalysts can extract the photogenerated holes and enhance the robustness of semiconductors [36,37]. These results imply that CuO is an efficient alternative to Pt for SrTiO$_3$.

In Figure 4a, it shows that the amount of H$_2$ evolution gradually increases from 166 μmol to 259 μmol with increasing the CuO content from 0.5 wt.% to 1.5 wt.%. However, followed with further increasing the CuO content from 1.5 to 2.5 wt.%, the amount of H$_2$ evolution drops rapidly from 259 μmol to 200 μmol. From these results, it can be seen that an excess amount of CuO (beyond the optimum loading) increases the probability of a recombination reaction, leading to a decrease in the photocatalytic hydrogen evolution activity. Therefore, the optimum loading amount of CuO cocatalyst in SrTiO$_3$ photocatalyst is to be 1.5 wt.% where the CuO-SrTiO$_3$ photocatalyst shows the highest photocatalytic activity.

### 2.3. Effects of Type and Concentration of Electron Donors on Hydrogen Evolution over CuO-SrTiO$_3$

It is known that electron donors are usually added into the reaction solution in order to improve the photocatalytic activity of photocatalyst by reducing the recombination of e$^-$/h$^+$ pairs. 60 mg of 1.5% CuO-SrTiO$_3$ photocatalyst is added into 60 mL of a solution containing various electron donors such as methanol (MeOH), glycerol (C$_3$H$_5$(OH)$_3$), ethylenediaminetetraacetic acid (EDTA) or triethanolamine (TEOA) to investigate the effects of electron donors on hydrogen evolution over CuO-SrTiO$_3$.

As shown in Figure 5, the photocatalytic hydrogen evolution activity over the 1.5% CuO-SrTiO$_3$ photocatalyst by adding different electron donors is in the following order: methanol > EDTA > glycerol > TEOA. Very clearly, compared to the other electron donors studied, methanol is the most effective electron donor for the CuO-SrTiO$_3$ photocatalyst. This is possibly due to its stronger ability of donating electrons to scavenge the valence band holes, which can effectively prevent photo-generated charge recombination. In the case of methanol as an electron donor, the amount of hydrogen evolution increases with enhancing concentration of methanol up to 60 vol.%. Beyond 60 vol.%, the hydrogen evolution rate increases little. An explanation of this result is that the reactions at the interface dominate the whole process of hydrogen evolution. Therefore, the rate of hydrogen evolution is limited by the amount of reactive species produced by methanol from the solution to the surface of the photocatalyst. When the concentration of methanol is lower than 60 vol.%, the adsorption of methanol on the surface of photocatalyst cannot saturate. Thus, the amount of reactive species does not yet reach maximum. When the concentration of methanol is beyond 60 vol.%, more methanol molecules are adsorbed on the surface of the photocatalyst. However, the relative amount of reactive species on the surface of the photocatalyst could not increase because the intensity of light and the amount of photocatalyst remain constant. Therefore, amount of hydrogen evolution enhances little followed with the increased amount of methanol. The most suitable concentration of methanol for hydrogen production

over the 1.5% CuO-SrTiO$_3$ photocatalyst is 60 vol.%. Based on the above results, 60 vol.% of methanol aqueous solution is used in following investigations.

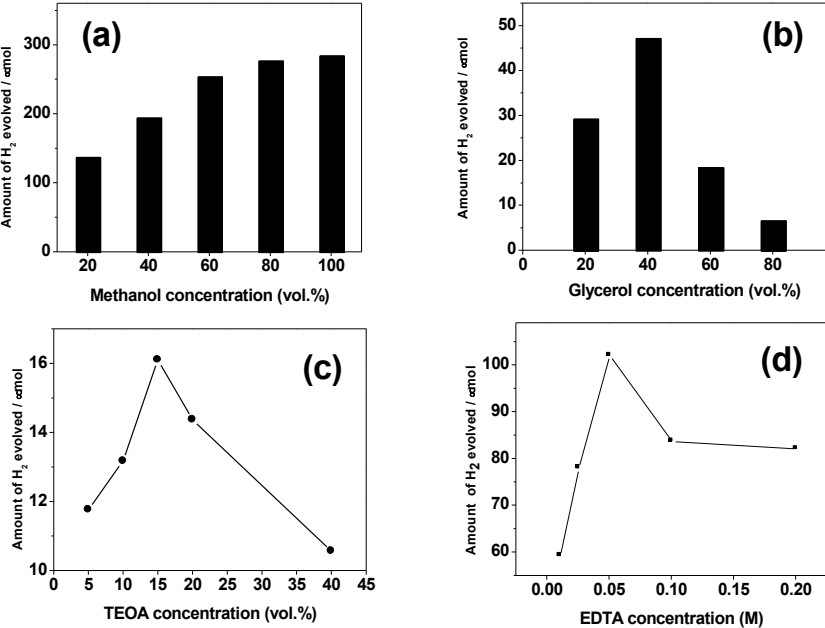

**Figure 5.** The effects of type and concentration of electron donors on the photocatalytic hydrogen evolution activity over 1.5% CuO-SrTiO$_3$: (**a**) methanol, (**b**) glycerol, (**c**) TEOA and (**d**) EDTA. The amount of catalyst: 60 mg; volume of reaction solution: 60 mL; reaction temperature: 25 °C; irradiation time: 2 h.

### 2.4. Effect of Reaction Temperature on Hydrogen Evolution over CuO-SrTiO$_3$

In order to determine the other suitable conditions for hydrogen production from the photocatalytic water splitting using methanol as the electron donor, the effect of reaction temperature on hydrogen evolution over CuO-SrTiO$_3$ was studied and the results are shown in Figure 6.

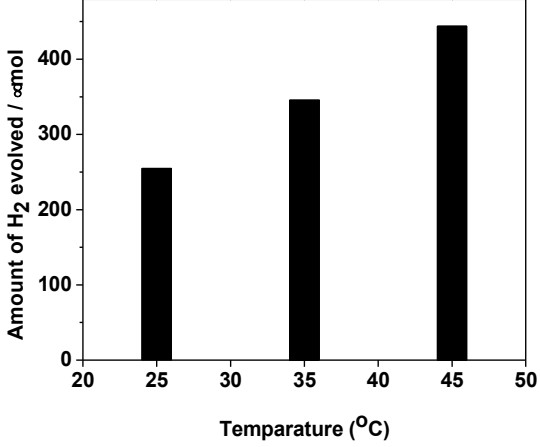

**Figure 6.** The effect of reaction temperature on photocatalytic hydrogen evolution activity over 1.5% CuO-SrTiO$_3$ photocatalyst. Catalyst: 60 mg; amount of methanol: 60 mL, 60 vol.%; irradiation time: 2 h.

It can be seen that the reaction temperature is a significant factor for the photocatalytic hydrogen evolution over the CuO-SrTiO$_3$ photocatalyst from the methanol aqueous solution. The results in Figure 6 show that the photocatalytic activity for hydrogen evolution is increased when increasing the reaction temperature. The amount of hydrogen evolution can reach 444 µmol at 45 °C, which is 1.75 times higher than that at 25 °C. According to the

reports of Puangpetcha and Korzhak [31,38], the effect of temperature on the photocatalytic hydrogen evolution reaction can be related to the thermal activation energy and desorption of the oxidation products from the sacrificial reagent (methanol).

### 2.5. Effect of Amount of Photocatalyst on Hydrogen Evolution

Figure 7 shows the effect of the amount of the 1.5% CuO-SrTiO$_3$ photocatalyst on the photocatalytic hydrogen evolution activity.

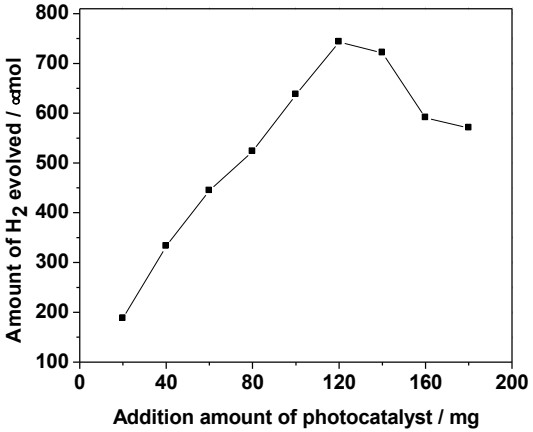

**Figure 7.** Effect of the addition amount of the 1.5% CuO-SrTiO$_3$ photocatalyst on the photocatalytic activity for hydrogen evolution. The amount of methanol: 60 mL, 60 vol.%; reaction temperature: 45 °C; irradiation time: 2 h.

The result in Figure 7 shows that the amount of hydrogen evolution increases from 187 μmol to 742 μmol when the addition amount of the 1.5% CuO-SrTiO$_3$ is enhanced from 20 mg to 120 mg. In contrast, the amount of hydrogen evolution decreases from 742 μmol to 570 μmol with further increasing the amounts of the 1.5% CuO-SrTiO$_3$ from 120 mg to 180 mg. Therefore, the maximum amount of hydrogen evolution is obtained when the addition amount of 1.5% CuO-SrTiO$_3$ photocatalyst is 120 mg per 60 mL (2 g/L) in the present system. It could be explained that at a low concentration of photocatalyst, the photocatalytic reaction is mainly governed by active sites which are available for absorption of light and adsorption of reactant. The active sites are increased with increment of the concentration of the photocatalyst. However, when the concentration of the photocatalyst is above the optimum level, the reaction system becomes turbid, and UV light is greatly scattered by the suspended photocatalyst. Therefore, the transmission of UV light in suspension is greatly inhibited, and results in a sharp decrement in photocatalytic activity for hydrogen evolution.

### 2.6. Photocatalytic Hydrogen Evolution Activity of 1.5% CuO-SrTiO$_3$

In order to evaluate photocatalytic hydrogen evolution activity of the 1.5% CuO-SrTiO$_3$ photocatalyst from methanol aqueous solution, the relationship between amount of hydrogen evolved and irradiation time is investigated under optimum conditions for prolonged irradiation time, with the pure SrTiO$_3$ as comparison. The experimental results are shown in Figure 8.

As illustrated in Figure 8, the amount of hydrogen evolution almost linearly increases followed with the prolonged irradiation time. After irradiation for 48 h, the amount of hydrogen evolution over 1.5% CuO-SrTiO$_3$ is 13.31 mmol, whereas it is only 0.08 mmol for pure SrTiO$_3$. The results presented above clearly indicate that loading CuO on the surface of SrTiO$_3$ can dramatically enhance the photocatalytic activity of SrTiO$_3$, and the CuO-SrTiO$_3$ photocatalyst is highly stable and active for the reduction of H$_2$O to H$_2$ in the presence of methanol.

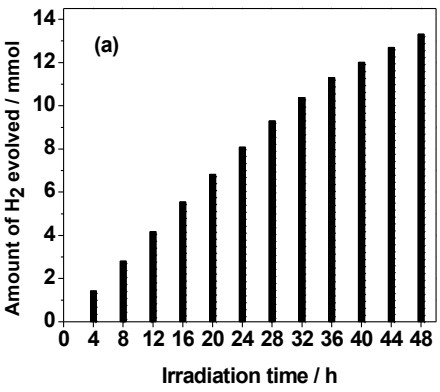 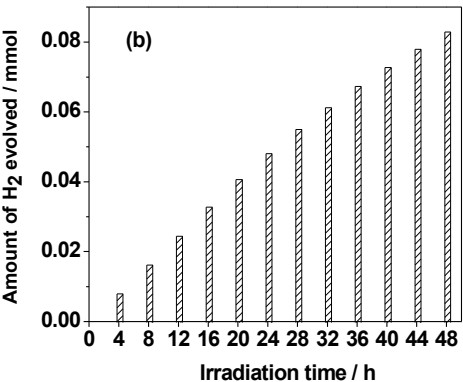

**Figure 8.** Time course of hydrogen evolution over 1.5% CuO-SrTiO$_3$ (**a**) and pure SrTiO$_3$ (**b**). The amount of photocatalyst: 120 mg; the concentration of methanol: 60 mL, 60 vol.%; reaction temperature: 45 °C.

The specific surface areas of pure SrTiO$_3$ and the 1.5 wt.% CuO-SrTiO$_3$ are 16.2 and 15.5 m$^2$g$^{-1}$, respectively. Very clearly, loading CuO on SrTiO$_3$ photocatalyst cannot change the specific surface area of SrTiO$_3$. Thus, the enhancement of hydrogen production by loading CuO on SrTiO$_3$ photocatalyst could not originate from the specific surface area of photocatalyst.

The UV–vis diffuse reflectance spectra of pure SrTiO$_3$ and 1.5% CuO-SrTiO$_3$ samples are shown in Figure 9. The absorption bands of the 1.5% CuO-SrTiO$_3$ sample are not shifted compared to that of pure SrTiO$_3$. Both of the samples exhibit an absorption band around 384 nm corresponding to band gap energy of 3.22 eV calculated from the formula E$_g$ = 1240/λ [39]. Obviously, loading CuO on the surface of SrTiO$_3$ does not change the band gap energy of the SrTiO$_3$ semiconductor. The same result has also been reported in literature [40] for the case of the CuO-loaded SrTiO$_3$ photocatalyst.

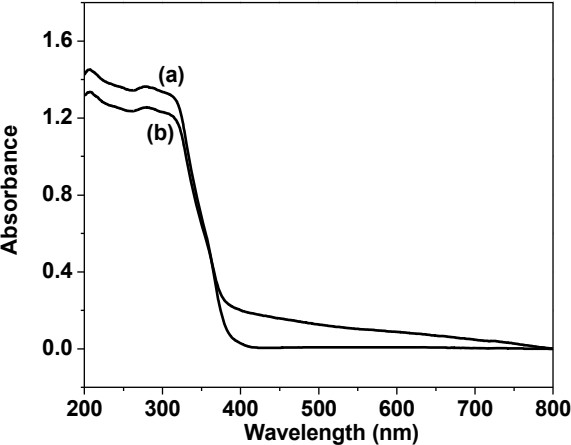

**Figure 9.** Diffuse reflectance spectra of pure SrTiO$_3$ (a) and 1.5 wt.% CuO-SrTiO$_3$ (b).

Moreover, the PL spectra of pure SrTiO$_3$ and the 1.5% CuO-SrTiO$_3$ are measured and shown in Figure 10. In general, if the amounts of the photoproduced electrons resulting from the recombination of excited electrons/holes are increased, the PL intensity of the sample increases. Consequently, its photocatalytic activity decreases. Therefore, there exists a close relationship between the PL intensity and photocatalytic activity of photocatalyst. As illustrated in Figure 10, the PL intensity of the 1.5% CuO-SrTiO$_3$ is lower than that of pure SrTiO$_3$. It is indicated that there exists a photo-generated electron transfer from SrTiO$_3$ to CuO, which efficiently depresses the recombination of photoproduced electrons/holes in the photocatalyst.

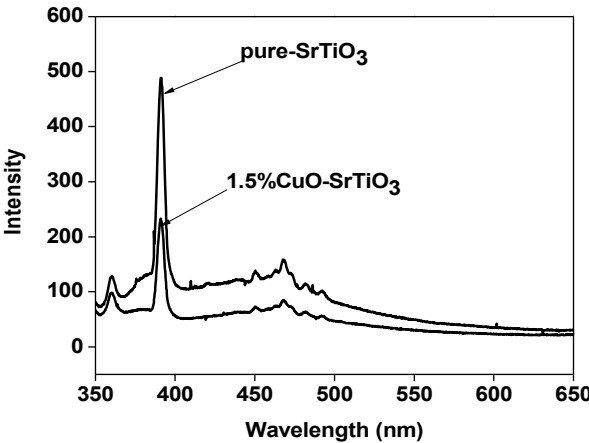

**Figure 10.** Photoluminescence (PL) spectra of pure SrTiO$_3$ and the 1.5% CuO-SrTiO$_3$.

Based on the above results and the literature reported [28,32], the great increment of the photocatalytic activity of 1.5% CuO-SrTiO$_3$ ought to be ascribed to the efficient separation of electron/hole pairs. In order to better understand the effect of loading CuO on photocatalytic hydrogen evolution over the CuO-SrTiO$_3$ photocatalyst from the methanol/water, a possible mechanism is developed. As can be seen in Figure 11, under UV light irradiation, the electrons in the valence band (VB) of the SrTiO$_3$ are transferred to the conduction band (CB) of the SrTiO$_3$. Due to the CB of CuO (+0.46 V) is situated below the CB of SrTiO$_3$ (−0.29 V), the excited electrons in the CB of SrTiO$_3$ can rapidly be transferred to the CB of CuO. The H$^+$ transferred to the surface of CuO cocatalyst particles are reduced to H$_2$ by the photogenerated electrons. As the electrons are captured by CuO, the recombination of electrons and holes is depressed. The holes produced on the valence band are allowed to oxidize H$_2$O and methanol, which makes the photocatalytic reaction be able to continue.

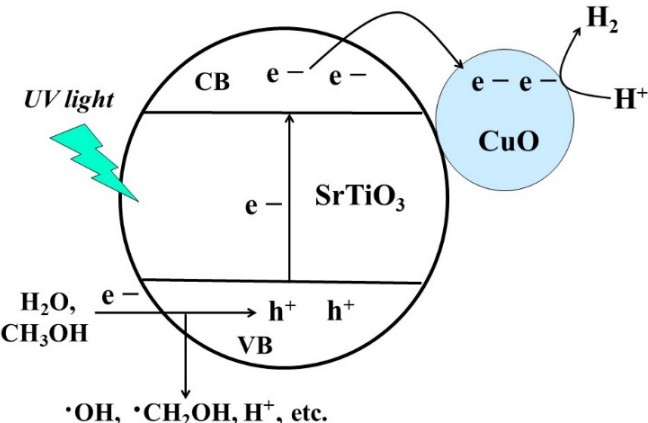

**Figure 11.** Scheme of photocatalytic hydrogen evolved reaction over the CuO-SrTiO$_3$ photocatalyst under UV light irradiation.

It is noticed that the potential of the CuO is not suitable for hydrogen evolved from water splitting because the CB level of CuO semiconductors (+0.46 V) is more positive than $E_{H_2O/H_2} = 0$ V (vs. NHE, pH = 0) [34]. However, once the photoproduced electrons are transferred to CuO, the Fermi level of CuO is raised which results in a more negative CB potential of CuO. In other words, the accumulation of excess electrons in CuO can cause a negative shift of the CB potential of CuO in the end. Thus, the reduction of water can be carried out at the CB of the CuO as shown in Figure 12.

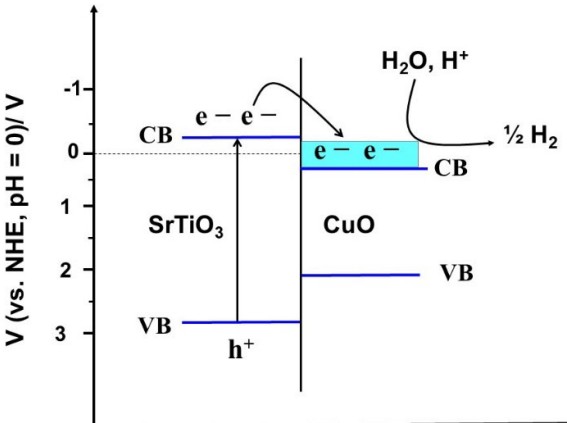

**Figure 12.** Schematic diagram of the energy band positions of SrTiO$_3$, CuO and the direction of electron transfer.

## 3. Materials and Methods

### 3.1. Materials

Strontium nitrate (Sr(NO$_3$)$_2$, Sigma-Aldrich, St. Louis, MO, USA), tetra-butyl titanate (TBOT Sigma-Aldrich), Cu(NO$_3$)$_2$ (Xilong Chemical Co., Ltd. Guangzhou, China). All other reagents were analytical grades and used without further purification.

### 3.2. Preparation of CuO-SrTiO$_3$ Photocatalyst

The SrTiO$_3$ nanoparticles loaded with various amounts of CuO (0.5, 1.0, 1.5, 2 and 2.5 wt.%) were prepared by impregnation method. A typical experimental procedure of 1.5 wt.% CuO-loaded SrTiO$_3$ was described as followed: 200 mg of the SrTiO$_3$ photocatalyst and 3.75 mL of 0.01 mol·L$^{-1}$ Cu(NO$_3$)$_2$ solution were added into 10 mL distilled water. Afterwards, the suspension solution was stirred at room temperature for 4 h. Finally, the sample was dried at 80 °C for 12 h and calcined at 450 °C for 4 h. The other CuO-loaded SrTiO$_3$ samples were prepared using similar procedures. The obtained products are denoted as $x$% CuO-SrTiO$_3$ ($x$% is weight per cent of CuO).

### 3.3. Characterization Techniques

The structure and crystalline phase of the samples were characterized by Rigaku D/max 2550 VB/PC X-ray diffractometer (XRD) with Cu K$\alpha$ radiation ($\lambda$ = 0.154056 nm) at 40 kV and 40 mA (Tokyo, Japan). The microstructure and morphology of the samples were analyzed by JEOL, JEM-200CX transmission electron microscope (TEM) with 200 kV accelerating voltage (Tokyo, Japan). The X-ray photoelectron spectra (XPS) were recorded on a PHI 5000 Versaprobe spectrometer (ULVAC-PHI, Inc., Kanagawa, Japan). The energy dispersive X-ray spectrocopy (EDX) was taken with a JEOL JSM-6360LV electron microscopy (Tokyo, Japan). Fourier transform infrared spectra (FT-IR) were recorded on a Bruker Vector 22 spectrometer (resolution 4 cm$^{-1}$, Yokohama, Japan), the samples being pressed into disks with KBr. The solid diffusion reflectance UV–vis spectra (DRS) were recorded on Unico UV-2102 PCS spectrometer (Tokyo, Japan). The photoluminescence spectra (PL) were recorded with a Shimadzu RF-5301PC fluorescence spectrometer (Tokyo, Japan). The N$_2$ adsorption and desorption isotherms were measured on a Micromeritics ASAP-2020 nitrogen adsorption apparatus (Norcross, GA, USA).

### 3.4. Photocatalytic Water Splitting Experiments of CuO-SrTiO$_3$ Photocatalyst

The photocatalytic reaction was carried out in a gas-closed system with a reactor made of quartz. In a typical experiment, 60 mg or 120 mg of the CuO-loaded SrTiO$_3$ photocatalyst was added into 60 mL of the reaction solution containing various electron donors. Before photo-irradiation, the mixed solution was sonicated for 3 min, and then the suspended solution was bubbled with highly pure N$_2$ gas (about 15 mL/min) for 30 min to remove

dissolved oxygen gas in the suspension. The suspension was irradiated by a 300 W high-pressure Hg lamp with cooling water was circulated through a cylindrical quartz jacket located around the UV light source. To maintain a constant reactor temperature, the photocatalytic reaction system was equipped with an electric fan and water with appropriate temperature was circulated around the quartz reactor. The amounts of $H_2$ evolution were analyzed by a gas chromatograph (GC-112A, molecular sieve 5A, TCD, Chongqing Gold Equipment Co., Ltd., Chongqing, China) and $N_2$ as a carrier gas.

## 4. Conclusions

Photocatalytic hydrogen evolution over CuO-loaded $SrTiO_3$ photocatalyst from aqueous solution containing various electron donors has been studied under UV irradiation. The effects of various factors, such as loading amount of CuO, type and concentration of electron donors, reaction temperature and concentration of the photocatalyst have been systematically investigated in detail. The results showed that methanol was the most effective electron donor for photocatalytic hydrogen evolution over the $CuO$-$SrTiO_3$ photocatalyst. The optimum reaction conditions for hydrogen evolution from methanol aqueous solution over $CuO$-$SrTiO_3$ were found out. That is, the loading amount of CuO was 1.5 wt.%, the concentration of $CuO$-$SrTiO_3$ photocatalyst was 2 g/L, the concentrating of methanol was 60 vol.%, and the reaction temperature was 45 °C. Importantly, it was found that the CuO loaded $SrTiO_3$ nanoparticles were highly active and stable photocatalyst for the photocatalytic hydrogen production from methanol aqueous solution.

**Author Contributions:** Conceptualization, X.T.M. and D.N.B.; methodology, H.D.C. and T.H.L.N.; software, T.T.L.N.; formal analysis, H.D.C. and T.H.L.N.; data curation, T.T.L.N. and X.T.M.; writing—original draft preparation, D.N.B. and V.K.P.; writing—review and editing, V.K.P. and T.K.N.T. All authors have read and agreed to the published version of the manuscript.

**Funding:** This work was supported the MOET of Vietnam Project No. B2020-TNA-12.

**Institutional Review Board Statement:** Not applicable.

**Informed Consent Statement:** Not applicable.

**Data Availability Statement:** All the data are available within the manuscript.

**Conflicts of Interest:** The authors declare no conflict of interest.

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
