# Peer review of "Effect of CuO Loading on the Photocatalytic Activity of SrTiO3 for Hydrogen Evolution"

_inorganics, doi:10.3390/inorganics10090130_

Round 1

Reviewer 1 Report

The authors report the preparation and catalytic performance of the CuO/SrTiO3 composite. The authors used CuO as a modifier to improve the photocatalytic performance in hydrogen evolution. According to the authors' results in the catalytic performance, CuO can enhance the catalytic performance of SrTiO3. And the authors also investigate the reason why CuO can play the important role in the evolution of hydrogen. However, the authors should consider the following points before the publication of this manuscript:

1) The lowest amount of hydrogen which can be measured by the authors was 1.45 umol, which corresponds to the hydrogen volume of  about 0.035 ml at the standard state. Can this amount be measured accurately by the GC? Moreover, the authors provide the molar amount of hydrogen with the number of significant digits being 5 (Tab. 1), and is it rational and necessary?

2)How about the catalytic activity of CuO? The authors provide the catalytic activity of CuO-SrTiO3 composite, the reference catalysts should include CuO.

3)How about the energy profile (CB/VB) of CuO and SrTiO3? The authors provided an interpretation, but the detailed energy profiles should be provided to judge the authors' interpretation.

According to above points, my recommendation is major revisions.

Author Response

Dear Editor and reviewers,

We would like to express our gratitude for the Editor and Reviewer’s efforts to improve the quality of our manuscript. We have tried our best to respond to all issues indicated in the review report sufficiently. In the revised version, we have highlighted the changes to our manuscript using the red color. The answers to the questions you raised are detailed here.

Reviewer 2 Report

In the manuscript, the authors have synthesized CuO- loaded SrTiO3 catalyst for efficient photocatalytic H2 evolution. Under optimun conditions, the best rate of H2 evolution of the CuO- loaded SrTiO3 catalyst is 5811 μmol h-1 g-1. The optimal H2 evolution rate of CuO-SrTiO3 is more than 60% of that of 2.0 wt.% Pt -loaded SrTiO3. However, several problems should be answered or solved for a major revision before this manuscript can be considered for the possible acceptance.

1. What is the particle size of the loaded CuO? The author claims that: “Based on the above results and the literature reported [29,31], the great increment of the photocatalytic activity of 1.5% CuO-SrTiO3 ought to be ascribed to the efficient separation of electron/hole pairs.” However, in the cited literature, the quantum size effects increase the band gap in CuO clusters and also shift the conduction band enough to permit transfer of photogenerated electrons. But in this work, the positions of CB for CuO (+0.46 V) and SrTiO3 (−0.29 V) are far apart. And the conduction band of CuO is positive (+0.46) and may not allow for direct transfer of electrons from CuO to H+ in the solution. The author needs to explain them.

2. The description of the mechanism in the manuscript needs to be clearer. Does CuO form heterojunctions with SrTiO3? According to the band positions of CuO and SrTiO3, the two materials will form a type I-heterojunction. However, type I-heterojunction is generally less efficient than type II-heterojunction. Because under light irradiation, the electrons and holes will accumulate at the CB and the VB levels of semiconductor B (CuO), respectively. Since both electrons and holes accumulate on the same semiconductor, the electron–hole pairs cannot be effectively separated for the type-I heterojunction photocatalyst. Moreover, a redox reaction takes place on the semiconductor with the lower redox potential, thereby significantly reducing the redox ability of the heterojunction photocatalyst. Therefore, the authors need to further clarify the mechanism.

3. The mechanim picture should also be optimized, which should include CB/VB levels of SrTiO3 and CuO. And draw the carrier transfer path in detail.

4. The photo-electrochemical should be measured to prove the efficiency of carrier separation for CuO-SrTiO3.

5. If possible, the XPS spectra of the CuO-SrTiO3 sample that subjected to prolonged illumination should be provided to further demonstrate the stability of the sample.

6. Some closely related papers can be referenced to boost the discussion :

Adv. Funct. Mater. 31, 2021, 2100553; ACS Appl. Energy Mater. 5, 2022, 6155

Author Response

(The authors gave the same response as above.)

Reviewer 3 Report

The manuscript entitled "Effect of CuO loading on the photocatalytic activity of SrTiO3 for hydrogen evolution" is an interesting research topic, this can be accepted after following revisions.

1.

2. The literature survey is incomplete, very important and closely related work is not addressed like https://doi.org/10.1016/j.jphotochem.2018.07.010, and 10.1016/j.jphotochem.2020.112886, etc.

3. The characterization of the material is very important, the weight % of copper used in the synthesis experiment can not be assumed to be the all present in the final catalyst's composition, therefore the exact amount of copper contents needs to be measured (EDX and XPS are performed but both provide information of surface contents only), the AAS/ICP for the total copper contents is recommended to followed for this.

4. For the catalytic activity; a comprehensive comparison of the literature reports (as highlighted just here above in the point #1 and many others) and the current results be presented. Only two references (# 29 and 31) are used by the authors for comparisons.

5. The catalytic mechanism needs appropriate references.

Author Response

(The authors gave the same response as above.)

Round 2

Reviewer 1 Report

The authors have revised the manuscript and responded to the comments. The revised manuscript is ready for publication except for some problems on the writing, and this manuscript can be accepted after the checking of the whole manuscript.

Author Response

Dear Editor and reviewers,

We would like to express our gratitude for the Editor and Reviewer’s efforts to improve the quality of our manuscript. We have tried our best to respond to all issues indicated in the review report sufficiently. In the revised version, we have highlighted the changes to our manuscript using the yellow color. The answers to the questions you raised are detailed here.

Reviewer 2 Report

I recommend this manuscript to be accepted after minor revision.

1.HRTEM images should be provided.

2.Some mistakes should be corrected, such as Fig. 4c.

3.Some closely related papers can be referenced to boost the discussion : Adv. Funct. Mater. 31, 2021, 2100553; ACS Appl. Energy Mater. 5, 2022, 6155

Author Response

(The authors gave the same response as above.)
